# Enterovirus Surveillance (EVSurv) in Germany

**DOI:** 10.3390/microorganisms9102005

**Published:** 2021-09-22

**Authors:** Kathrin Keeren, Sindy Böttcher, Sabine Diedrich

**Affiliations:** 1Secretary of the National Commission for Polio Eradication in Germany, Robert Koch Institute, 13353 Berlin, Germany; keerenk@rki.de; 2National Reference Centre for Poliomyelitis and Enteroviruses, Robert Koch Institute, 13353 Berlin, Germany; boettchers@rki.de

**Keywords:** Germany, enterovirus, surveillance, viral meningitis, aseptic meningitis, acute flaccid paralysis, AFP, poliovirus eradication

## Abstract

The major aim of the enterovirus surveillance (EVSurv) in Germany is to prove the absence of poliovirus circulation in the framework of the Global Polio Eradication Program (GPEI). Therefore, a free-of-charge enterovirus diagnostic is offered to all hospitals for patients with symptoms compatible with a polio infection. Within the quality proven laboratory network for enterovirus diagnostic (LaNED), stool and cerebrospinal fluid (CSF) samples from patients with suspected aseptic meningitis/encephalitis or acute flaccid paralysis (AFP) are screened for enterovirus (EV), typing is performed in all EV positive sample to exclude poliovirus infections. Since 2006, ≈200 hospitals from all 16 German federal states have participated annually. On average, 2500 samples (70% stool, 28% CSF) were tested every year. Overall, the majority of the patients studied are children <15 years. During the 15-year period, 53 different EV serotypes were detected. While EV-A71 was most frequently detected in infants, E30 dominated in older children and adults. Polioviruses were not detected. The German enterovirus surveillance allows monitoring of the circulation of clinically relevant serotypes resulting in continuous data about non-polio enterovirus epidemiology.

## 1. Introduction

Together with the partners within the network of the Global Polio Eradication Initiative (GPEI)—UNICEF, Rotary International and the US CDCs—WHO has succeeded in pushing back poliomyelitis to a large extent. Wild poliovirus (WPV) types 2 and 3 are eradicated, and only WPV 1 is still endemic in Pakistan and Afghanistan [1,2]. However, until the final steps of eradication are not achieved, polioviruses can be reimported into polio-free areas (e.g., Tajikistan [3], Republic of Congo [4], Syria [5], Iraq [6], Israel [7]). Therefore, an effective surveillance method such as surveillance of acute flaccid (AFP) paralysis, enteroviruses or environmental samples has to be maintained until final eradication to identify poliovirus circulation as early as possible to immediately start outbreak response [8].

In Germany, the last indigenous polio case occurred in 1990. After two importations in 1992, up to three vaccine-associated paralytic polio cases (VAPP) were registered during the following years. Therefore, in 1998 the vaccine strategy in Germany was switched from attenuated live virus (OPV) to inactivated virus (IPV). The national AFP surveillance was established in 1998. Since the results of the AFP surveillance were insufficient, an alternative system for the surveillance was implemented. Samples from hospitalised patients with suspected meningitis are tested for the presence of enteroviruses. As a result of the investigation of mainly faecal specimens, this system allows the monitoring of circulating enteroviruses and hence the absence of polioviruses in Germany.

Enteroviruses are members of the family *picornaviridae* and are globally widespread. More than 100 different types have been detected in humans and are now classified into four species, A–D [9]. During the last decade, re-emerging enterovirus types (e.g., enterovirus D68; EV-D68 and enterovirus A71; EV-A71) were associated with outbreaks of severe respiratory (USA 2014 [10]) and central nervous system disorders (Spain 2016 [11]). A rare complication of enterovirus infections is acute flaccid paralysis which may also result from acute flaccid myelitis (AFM) and poses a public health threat. Furthermore, atypical courses of hand, foot and mouth disease (HFMD) mainly caused by coxsackievirus A6 were first described in 2008 (Spain [12], Finland [13]) and are now present all over the world [14,15,16].

We describe the results of the enterovirus surveillance (EVSurv) conducted in Germany between 2006 and 2020.

## 2. Materials and Methods

### 2.1. Rational for Enterovirus Surveillance in Germany

According to the WHO guidelines [17], AFP surveillance still represents the gold standard to prove polio-free status. As a supplementary system in high-risk areas, testing of sewage samples can be performed. Nevertheless, countries that have been polio-free for a long period may have problems fulfilling the WHO criteria for AFP surveillance or are unable to conduct nationwide environmental surveillance. During the final phase of poliovirus eradication, syndromic surveillance systems, including testing of different clinical samples (faecal, respiratory, CSF) from patients with symptoms compatible with a poliovirus infection, can be established. Most poliovirus infections remain asymptomatic, with paralytic poliomyelitis being present in less than 1% of people infected with polioviruses, but more patients develop symptoms of aseptic meningitis (2–5%). We, therefore, aimed to establish a surveillance system including both syndromes to increase the probability of detecting poliovirus infections. Furthermore, the system contributes to the etiological clarification of aseptic meningitis or encephalitis and non-polio AFP what could help to improve patient management and increases our knowledge on the circulation of clinically relevant enterovirus types.

### 2.2. Establishment of a Laboratory Network for Enterovirus Diagnostic in Germany (LaNED)

During the period of AFP surveillance (1997–2010), the administrative work (case registration, weekly request for AFP cases, newsletter) was conducted by the Governmental Institute of Public Health of Lower Saxony (NLGA). According to WHO recommendation, testing of stool samples has to be performed by a WHO accredited lab. Therefore, samples were tested solely by the National Reference Centre for Poliomyelitis and Enterovirus (NRZ PE) at the Robert Koch Institute, Berlin (RKI).

Due to the expected increase in samples of about 3000 per year to be analysed when including patients with suspected meningitis/encephalitis, other labs were requested to take part in a laboratory network for enterovirus detection. Required techniques for participation were (I) pan-enterovirus RT-PCR, (II) virus isolation in cell culture on at least RD-A cells and (III) typing by neutralisation test/immunofluorescence and/or molecular methods. Individual lab performance is proven by a mandatory biannual proficiency test offered by NRZ PE in cooperation with INSTAND e.V. [18]. This enables labs to use the protocols best suitable to their individual lab arrangements and guarantees high quality of enterovirus testing within the network.

Initially, 14 labs joined the network and signed a cooperation agreement with RKI, which took over administrative responsibilities in 2010. Labs were requested to report results and pseudonymised patient data on a weekly basis to the RKI. Clinical samples, as well as isolates that could not be typed, were sent to the NRZ PE for the exclusion of polioviruses. All network labs received an expense allowance. Currently, 12 labs, including five universities, five governmental and two private labs (Figure 1), participated in the network.

Prior to the launch of EVSurv, the Ethics Commission of the Medical Association of Lower Saxony confirmed that this is not medical research on humans with epidemiological and personal data. Therefore, an ethics vote was not required.

Data analysis was performed in Excel (percentages and means) and GraphPad PRISM (odds ratio).

### 2.3. Recruitment of Hospitals

In Germany, infections with non-polio-enteroviruses (NPEV) are not notifiable at the federal level, and enterovirus testing is not paid for by health insurance. In order to offer an incentive to clinicians to send in stool samples, testing is free of charge. In 2005, 679 hospitals (383 pediatric and 296 neurological wards) were informed about the newly established system and requested to send adequate samples. Two hundred and eighty-seven hospitals agreed to participate on a voluntary basis. The number of contributing participants remained almost stable, with an average of 200 (166–238) per year.

### 2.4. Performance/Workflow of Enterovirus-Surveillance (EVSurv)

In order to prove the polio-free status of Germany, clinicians are requested to send in preferentially stool samples or CSF of patients with suspected aseptic meningitis/encephalitis or AFP for enterovirus diagnostic to one of the network labs. The presence of key symptoms (fever, headache, nausea/vomiting, nuchal rigidity, altered consciousness, paralysis of the limbs) is requested. The date of onset of symptoms and collection date as well as additional data on vaccination and travel history, are also requested within the form. The network lab receiving the samples performs enterovirus diagnostic, reports the results to the clinician and forwards results (including typing) in a regular manner to RKI (Figure 1). In accordance with the data protection law, personal data are pseudonymised (month and year of birth, sex and first three digits of residence). These data are submitted to a database and checked for plausibility.

Nationwide data are updated daily, and detailed queries can be run on https://evsurv.rki.de/ (accessed on 28 July 2021). A summarised version can be found at: http://www.rki.de/DE/Content/Institut/OrgEinheiten/Abt1/FG15/Polio-Kommission_Geschaeftsstelle/Geschaeftsstelle_inhalt.html?nn=2389946#doc3470414bodyText5 (accessed on 28 July 2021). Aggregated EVSurv results are reported to WHO on a regular basis.

### 2.5. Enterovirus Detection Workflow at the National Reference Centre for Poliomyelitis and Enteroviruses (NRZ PE)

For molecular analyses, RNA extraction and pan-enterovirus RT-PCR are performed as described recently [19]. Sequencing of the 5′NCR amplicon and query using BLAST [20,21] and the RIVM Typing tool [22] results in the identification of enterovirus species *Enterovirus A*, *Enterovirus B*, *Enterovirus C* and *Enterovirus D*. Since 2011, subsequent performance of species-specific RT-PCR assays covers the entire or partial VP1 region [23,24,25]. In the case of an *Enterovirus C* result, two RT-PCR assays are performed: Pan-Polio-VP1 [26] and a PCR, targeting 5′ half of the VP1 region [27]. Sequencing of the amplicons using second-round PCR primers and comparison with reference strains using BLAST and the RIVM enterovirus typing tool identifies the enterovirus type. PCR negative samples are tested for inhibition using MS2 phages.

For virus isolation, enterovirus RT-PCR positive stool suspensions and CSF samples, as well as virus isolates, sent in from the network labs are inoculated on at least one cell line (RD-A, CaCo2, Hep-2, L20B). All samples from patients with acute flaccid paralysis as well as all inhibited samples are inoculated immediately. All cell culture supernatants are passaged two times onto fresh cells, and cytopathic effects (CPE) are documented. Virus isolates are typed by neutralisation assay using type-specific antibodies and/or RIVM antibody pools. In the case of CPE under neutralisation, the inoculum is harvested, passaged once, and VP1 sequencing is performed to identify the reason for the failure of neutralisation (e.g., mixed infection).

## 3. Results

In total, 37031 samples were sent in from all 16 federal states from 2006 to 2020, with about 2500 (1307–3455) samples tested annually (Table 1). In 2020, about half as many samples were sent to a LaNED as in previous years. Specimens included 70% stool and 28% CSF samples (2% other material, e.g., serum, respiratory sample or not specified). The proportion of stool samples could be increased over time (2006: 58% stool; 2019: 79% stool). The majority of samples tested were from children <15 years (84%, median four years). Overall, enteroviruses were detected in approximately 25% of the samples (Table 1).

The enterovirus detection rate in stool samples was significantly higher (28.7%) compared to CSF (16.9%). In 2020, the positive rate (3%) was substantially lower than in previous years (20–38%).

More samples from males (57%) were sent in for analysis than from females (42%). Moreover, the positivity rate of samples from males (28.7%, odds ratio (OR): 1.385) was higher than from females (22.1%, OR: 0.733).

Analysis of age groups (>1 year, 1–4 years, 5–9 years and 10–14 years) revealed that the proportion of samples (27.1%), as well as the rate of enterovirus positive specimens (34.8%), were highest in the five- to nine-year-old age group. Since 2014, however, the number of samples from children older than five years of age has declined, while the number of samples from children younger than five years has remained stable or even increased. Figure 2 shows the change in the age distribution of investigated and positive detected samples over time.

For all enterovirus positive samples, further typing was performed. Overall, 83% of the enterovirus strains detected could be assigned at least to species level. By using molecular and virological methods, enterovirus typing improved over the years (94% typed in 2019 vs. 58% in 2006). The reasons for missing typing results in recent years are, e.g., limited sample volume or initially positive test results that could not be confirmed by the NRZ PE. Thus far, a total of 53 different types have been detected (Table 2). Polioviruses were not detected during the entire time period.

Figure 3 shows the seasonality of enterovirus infections with a peak from July through September (representing 61% of all enterovirus positive samples) combined with the proportion of echovirus 30 (E30) being the most prominent type over the years (*n* = 2231). Detection of E30 reached peak values in 2008, 2013 and 2018, supporting a five-year pattern that has been suggested recently [19]. E30 was detected in 72% of all enterovirus-positive samples in 2008 and in 61% of all enterovirus-positive samples in 2013. In 2018, however, E30 was also the most prominent serotype detected in 31% of the positive enterovirus samples, but to a far lesser extent than in previous epidemic years.

Other frequently detected types were echovirus 6 (E6, *n* = 786), enterovirus A71 (EV-A71, *n* = 538) as well as coxsackievirus B5 (CVB5; *n* = 406) and echovirus 11 (E11; *n* = 382). Figure 4 shows the yearly occurrence for these most common types. Except for EV-A71 upsurging every three years, no further special patterns became obvious.

Enterovirus A71 is responsible for large outbreaks of HFMD in South East Asia countries. In Europe, it is mainly associated with CNS symptoms. In order to survey the circulation of the different subgenotypes, molecular typing of EV-A71 is conducted by sequencing of VP1 region. Between 2007 and 2020, different EV-A71 different sub-genotypes could be identified (B5, C1, C2, C4), and also a recombinant C1-like strain emerging in 2015 (Figure 5).

As shown in Table 1, the majority of the samples analysed within the EVSurv are from patients with suspected aseptic meningitis/encephalitis. Since 2006, AFP was mentioned in 2.2% (*n* = 29–79 annually) of the patients, with a small proportion being infected with non-polio enteroviruses (three–seven annually). As the EVSurv does not document any MRI findings, the proportion of patients with evidence of acute flaccid myelitis (AFM) is not known. A total of 20 different enterovirus types was identified in AFP patients (Table 2). EV-A71 was detected most frequently. EV-D68, which is discussed to be associated with AFP/AFM cases in the USA, was detected once.

## 4. Discussion

The gold standard of polio surveillance is testing stools from patients with acute flaccid paralysis (AFP) for poliovirus. The AFP surveillance is used to achieve or confirm polio-free status in many countries [28,29,30,31,32]. However, polio-free countries are challenged by the sensitivity criteria (at least one AFP case per 100000 children under the age of 15 years per year) [33]. Therefore, alternative surveillance systems (environmental or enterovirus surveillance) are recommended [34,35]. In the United States, the “National Enterovirus Surveillance System” (NESS) is performed, which is a passive, voluntary surveillance system that has monitored laboratory detections of enteroviruses and parechoviruses since the 1960s [36,37]. Likewise, Germany decided on establishing a laboratory network for enterovirus surveillance (LaNED). Data from monitoring the circulation of enteroviruses will be increasingly important in the post-polio eradication era to detect poliovirus and other enteroviruses, causing potentially severe infections. For example, starting from 2014, there were outbreaks of enterovirus D68 associated with a severe respiratory infection and polio-like acute flaccid myelitis (AFM) in North America and several European countries [38,39,40,41,42,43,44,45,46]. There was also an upsurge in reported AFP cases in Germany in 2016 (total *n* = 78), revealing at least 16 cases presenting with AFM [47]. Enterovirus D68 is far more likely to be detected in respiratory specimens than in stool specimens which are the desired specimen type within EVSurv. As a result, the additional testing of a respiratory sample from patients with AFP/AFM was advised in 2017 [25,48,49]. As shown recently, several new enterovirus types assigned to species enterovirus C were identified in respiratory samples [50,51,52,53,54].

The EVSurv described here is available to all pediatric and neurologic hospitals free of charge and offers quality-tested enterovirus diagnostics for patients with polio compatible symptoms (viral meningitis, acute flaccid paralysis). Since enterovirus infections are not notifiable at the federal level in Germany (except for poliomyelitis), the cases recorded in the EVSurv are based on the voluntary contribution by the participating clinicians. EVSurv has received samples and data from approximately 200 hospital wards per year from all 16 federal states since 2006. Therefore, a stable database can be assumed. Since 2006, a total of 37031 samples have been analysed within the laboratory network. All participating network labs perform EV screening assays of their choice and are instructed to report their results on a weekly basis. To ensure the quality of diagnostic, all labs have to prove their performance by biannual proficiency tests applying molecular and virological methods. These proficiency tests are organised by NRZ PE in cooperation with INSTAND e.V. [18]. The diversity of used methods increases the chance to find rarely detected or new strains. In general, molecular typing has increased in recent years because it saves time compared to cell culture and has become affordable [55]. Nevertheless, typing of enteroviruses directly from clinical material remains challenging due to low viral load, e.g., in CSF [56]. Moreover, virological methods serve as a complementary tool to molecular methods. Virus isolates allow deeper insights into virus biology (antigenicity, growth kinetics, conduction of type-specific seroprevalence studies). In addition, virus isolation could also serve as a backup tool to identify variants that might be missed due to changes in the primer binding region [50]. Therefore, virus isolation has been the standard method for poliovirus detection recommended by GPEI for many years. In the context of poliovirus containment, molecular methods for direct detection of PV from stool extracts without using cell culture were developed [57,58,59]. As long as polio is not eradicated worldwide, there is a risk for importation into polio-free regions [60,61,62,63,64,65]. Therefore, a timely and valid diagnostic is important for adequate outbreak detection and control [3,66,67,68]. Since the detection of poliovirus is notifiable in Germany and a national action plan is in place, even a delayed report from a network lab to the surveillance system EVSurv would have no impact on an immediate start of the outbreak response.

Within the EVSurv, mainly stool samples or CSF from patients with polio-compatible symptoms, i.e., suspected aseptic meningitis/encephalitis or AFP, are tested for the presence of enteroviruses. Since 2006, approximately 25% of the samples have been enterovirus positive each year, which is in line with data from other countries [69,70,71,72,73], although data vary due to differences in study design, case definitions, sample type and age groups considered [71,74,75]. In recent years annual sample numbers and overall positivity rate have slightly decreased in Germany. Availability and affordability of multiplex tests in hospitals may be a reason for the decline. Indeed, enterovirus positive specimens are still not routinely typed or forwarded to a LaNED, although timely molecular typing can not only rule out poliovirus infection but also clarify the suspicion of a nosocomial outbreak or a community-acquired infection [76]. In total, EVSurv data show the highest sample numbers and positivity rate for the age group between 5 and 9 years. However, since 2014, the portion of samples from patients below five years of age increased not only due to a slight increase in samples from infants but mainly due to a lack of samples from older children. Particularly low sample numbers and positivity rate were observed in 2020. Intensified hygiene, wearing masks and physical distancing, as well as school and day-care closures implemented due to the SARS-CoV-2 pandemic, are also effective in preventing other infections (including enteroviruses) [77,78]. In contrast, exceptionally high detection rates of enteroviruses with high sample numbers were documented in 2008 and 2013, mainly due to infections with E30. Large outbreaks worldwide have repeatedly been linked to E30 [79,80,81,82,83], which was shown to be the predominant type in meningitis cases caused by enteroviruses [84]. Consistent with our data from EVSurv, E30 is also associated with AFP cases, although to a lesser extent. Furthermore, the most common non-polio enterovirus associated with AFP and encephalitis to date is EV-A71 [84]. Our data support this observation. Molecular typing of EV-A71 routinely conducted for early identification of emerging strains can assist public health authorities and clinicians to intensify enterovirus diagnostics. Furthermore, sequencing of the VP1 region and the entire genome allows assignment to genotypes, sub-genotypes and lineages and enables analysis of molecular evolution and phylogeography [85,86,87,88]. After detection of EV-A71 sub-genotype C4 in Germany in 2011–2013, a novel EV-A71 C1-like strain was described in 2015 [23], first reported in a child with severe rhombencephalitis [89]. Subsequent detection of these recombinant strains all over Europe [11,42,90,91,92] and also Asia [93,94,95] and the USA [96] was shown. Sub-genogroup C1, C2 and C4 circulate in Germany, with C2 being the predominant one until 2016 [23,89]. Consistent with data from other countries [97], EV-A71 shows a circulation pattern in Germany. Depending on the geographic region, different types exhibit different patterns. Malaysia and Japan show, as Germany does, a three-year pattern, whereas EV-A71 infections annually circle in China [98]. Serotype-specific immunity influenced by altered viral properties (pathogenicity, antigenicity, or transmissibility) likely explain these patterns [99].

However, the majority of enterovirus infections (including poliovirus infections) remain asymptomatic or present with unspecific symptoms. Unfortunately, samples tested positive for enterovirus/rhinovirus irrespective of enterovirus specific symptoms in hospitals are not routinely typed. In the context of polio eradication, however, the exclusion of poliovirus is essential. Because it is known that 2–5% of poliovirus infections can lead to aseptic meningitis, EVSurv relies on these cases to identify possible polio infections. However, asymptomatic infections would not be identified within syndromic surveillance systems. This became obvious in 2013 when two cases of asymptomatic poliovirus infections in immunocompromised children from Middle Eastern countries were identified [100]. Refugee and migration movements pose another challenge to syndromic surveillance capabilities. Hence, in response to the 2013 outbreak of poliomyelitis caused by wild poliovirus type 1 in Syria, stool samples from refugees arriving in Germany from Syria were tested for poliovirus, resulting in the identification of 12 individuals who excreted vaccine polioviruses [60].

In conclusion, (I) the establishment of laboratory-based surveillance in addition to our current surveillance system could increase the sensitivity of detection of poliovirus infection; (II) the implementation of screenings or dedicated surveillance for poliovirus in patients with primary immunodeficiency diseases (PIDs) could also be useful [101]; (III) the flexibility to analyse additional samples in challenging situations, such as increased migration from countries struggling with poliovirus outbreaks or weakened health systems, is required.

Moreover, the SARS-CoV-2 pandemic showed the impact of wastewater-based epidemiology (WBE) as an additional surveillance system. With regard to poliovirus detection, WBE was demonstrated to function as an early warning system in the absence of clinical cases [102,103]. Environmental surveillance is being conducted in many countries, especially those suffering from or at high risk of WPV or cVDPV outbreaks [104]. However, other countries that have long been polio-free rely on the results of this method [105,106,107]. In addition, using sewage samples for next-generation sequencing, several pathogens can be analysed at once, and circulation patterns of enteroviruses or the emergence of new and recombinant enterovirus types can be monitored [108,109].

In order to further improve not only enterovirus monitoring and diagnostics but also communication between experts and focus capacities in the field of enteroviruses, the European non-polio enterovirus network (ENPEN) was established. The network already published data on surveillance and laboratory detection of non-polio enteroviruses in the EU [55,80,110,111,112].

## Figures and Tables

**Figure 1 microorganisms-09-02005-f001:**
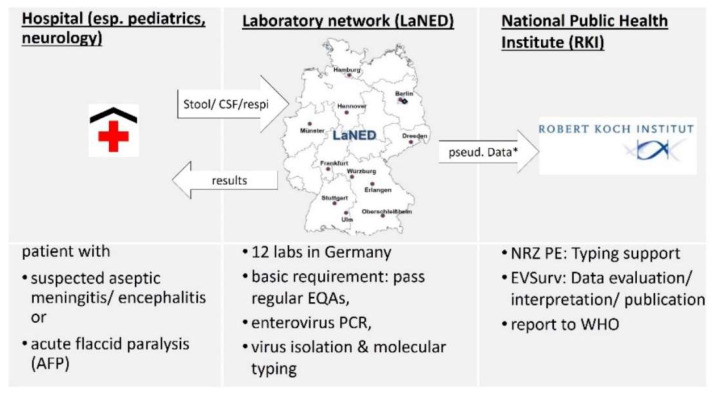
Workflow of EVSurv: * relevant pseudonymised data: sex, month/year of birth, first three digits of residence, symptoms, date of onset, polio vaccination status, material, travel history.

**Figure 2 microorganisms-09-02005-f002:**
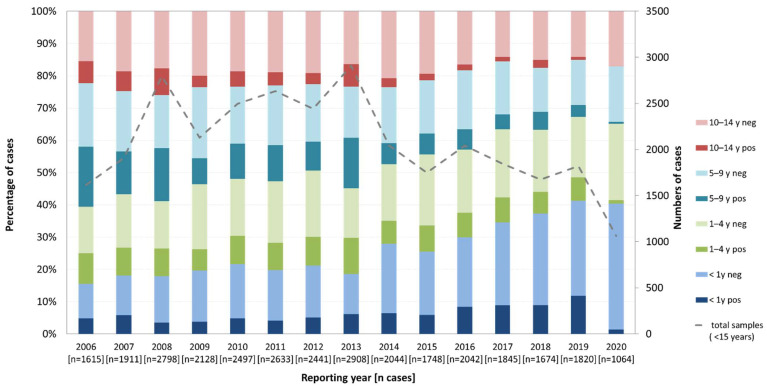
Numbers and proportion of enterovirus detections between 2006 and 2020 in children below 15 years of age and percentage in age groups (%).

**Figure 3 microorganisms-09-02005-f003:**
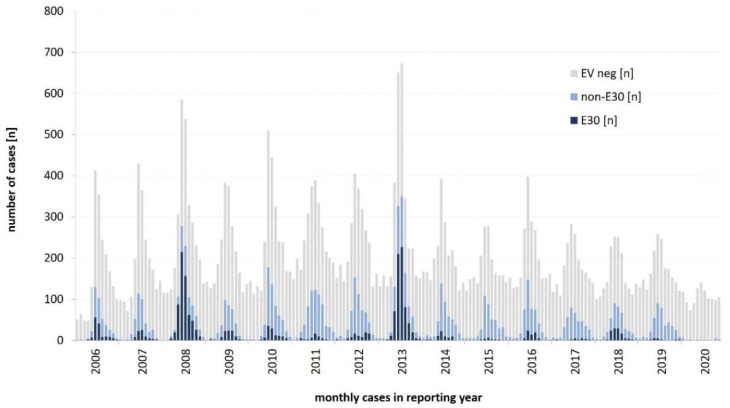
Seasonality of enteroviruses and proportion of E30 detections, monthly, 2006–2020.

**Figure 4 microorganisms-09-02005-f004:**
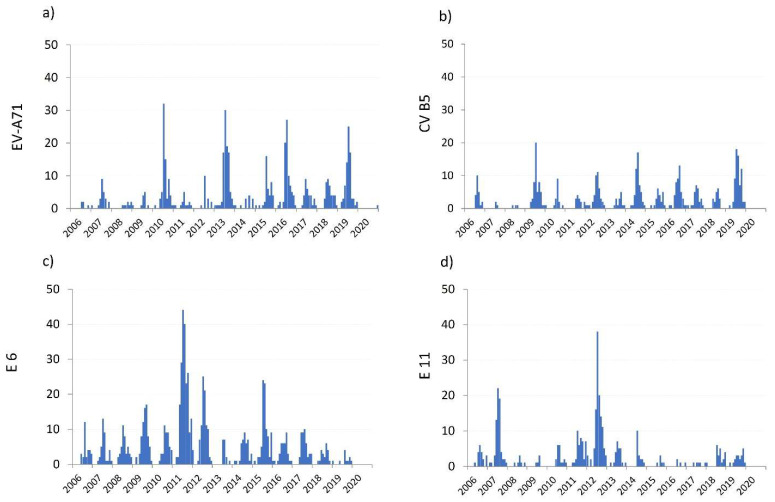
Pattern of the most prominent enterovirus types since 2006 (**a**) EV-A71: *n* = 538; (**b**) CV B5: *n* = 406; (**c**) E6: *n* = 786; (**d**) E 11: *n* = 382.

**Figure 5 microorganisms-09-02005-f005:**
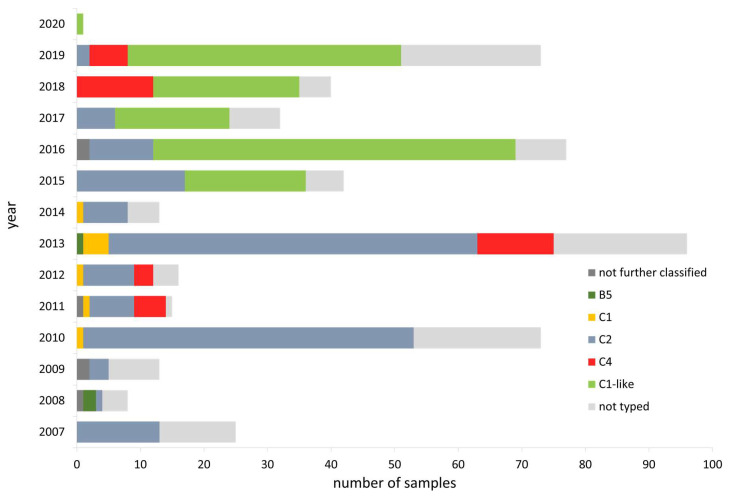
Number/proportion of EV-A71 sub-genogroups identified within the EVSurv between 2007 and 2020.

**Table 1 microorganisms-09-02005-t001:** Results of enterovirus surveillance (EVSurv) in Germany 2006–2020.

Year	Total [*n*]	Enterovirus Positive [*n*]	Enterovirus Positive [%]	Typed, Incl. NPEV [%]	Thereof AFP	
					Total [*n*]	Enterovirus Positive [*n*]
2006	1964	707	36%	58%	52	3
2007	2235	685	31%	57%	54	3
2008	3216	1132	35%	83%	66	3
2009	2556	513	20%	78%	62	6
2010	2947	793	27%	75%	54	6
2011	3074	775	25%	88%	61	6
2012	2926	700	24%	87%	50	7
2013	3455	1297	38%	91%	76	7
2014	2508	518	21%	95%	46	3
2015	2172	448	21%	94%	56	4
2016	2445	536	22%	92%	78	6
2017	2188	450	21%	92%	54	6
2018	1964	438	22%	92%	29	4
2019	2087	447	21%	94%	47	3
2020	1307	34	3%	91%	31	0
total	37031	9472			816	67

**Table 2 microorganisms-09-02005-t002:** Number of typed enteroviruses (Numbers in brackets indicate AFP cases with known enterovirus serotype). (* = without further subtyping.) Polioviruses were not detected.

Enterovirus A		Enterovirus B				Enterovirus C		Enterovirus D	
Serotype	[*n*]	Serotype	[*n*]	Serotype	[*n*]	Serotype	[*n*]	Serotype	[*n*]
CVA2	97 (2)	CVA9	214 (2)	E12	3	CVA1	2	EV-D68	11 (1)
CVA4	66 (4)	CVB1	69	E13	70	CVA19	4		
CVA5	34 (3)	CVB2	125 (2)	E14	30	CVA20	1		
CVA6	158 (1)	CVB3	137 (1)	E15	16	CVA22	7		
CVA7	9	CVB4	196 (3)	E16	27	CVA24	2		
CVA8	15 (1)	CVB5	406	E17	4	EV-C99	1		
CVA10	88 (2)	CVB6	5	E18	313 (4)	EV-C105	1		
CVA12	2	E1	2	E19	8				
CVA14	4	E2	14	E20	14				
CVA16	62 (2)	E3	54	E21	37				
EV-A71	538 (6)	E4	80	E24	13				
EV-A76	2	E5	47	E25	204 (1)				
		E6	786 (4)	E26	2				
		E7	105 (1)	E27	4				
		E9	288 (1)	E30	2231 (4)				
		E11	382 (1)	E31	9				
				E33	12				
EV-A *	45	EV-B *			177	EV-C *	3	EV-D *	0
EV-A, *n* = 1120		EV-B, *n* = 6084				EV-C, *n* = 21		EV-D, *n* = 11	

## Data Availability

The data presented in this study are available in this article.

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
