# Peer review of "Enterovirus Surveillance (EVSurv) in Germany"

_microorganisms, 2021, doi:10.3390/microorganisms9102005_

Round 1

Reviewer 1 Report

This manuscript is well written and I understood that enterovirus surveillance in Germany works efficiently.   The system provides a free of charge enterovirus diagnostic (on average, 2,500 samples per year!) at quality proven laboratory, showing polio-free.   To improve the syndromic surveillance, they included not only acute flaccid paralysis, but also aseptic meningitis/encephalitis cases.   They perform a mandatory biannual proficiency test, which require RT-PCR, virus isolation and typing by neutralization test/immunofluorescence and/or molecular methods.   I think such an occasion is necessary to keep the lab above the standard level.   I agree that “virus isolates allow deeper insights into virus biology (antigenicity, growth kinetics, conduction of type specific seroprevalence studies, variant investigation) (lines 256-259), although virus isolation has been replaced by molecular methods recently.   I hope the surveillance system is going to include even parechoviruses, which sometimes cause severe infectious diseases such as sepsis-like illness among young children, in the future.

Author Response

Dear Reviewer,

Thank you for reading our manuscript carefully and for your positive feedback.

Since the rationale of EVSurv is primarily the exclusion of poliovirus, parechovirus diagnostic is not performed. Nevertheless, some laboratories additionally include testing for parechoviruses (especially for newborns and infants).  

Sincerely,

Sabine Diedrich

Reviewer 2 Report

Please find attached my review.

Author Response

Dear Reviewer,

thank you for reading our manuscript carefully and for your positive feedback.

We would like to comment on your questions:

Question 1) Due to different geopolitics conflict in the world at this period (2006-2020), Germany as other European countries hosted many immigrants from different countries over the world, did the authors pay intention to the origin of the specimens?

Answer:

The data reported to EVSurv contain the first three digits of the patient's place of residence. The place of residence may be a refugee shelter. There is a free text section in the EVSurv form where travel history information can be provided, but it is not regularly used. We have attempted to address migrants in the Discussion section starting at line 325:

“Refugee and migration movements pose another challenge to syndromic surveillance capabilities. Hence, in response to the 2013 outbreak of poliomyelitis caused by wild poliovirus type 1 in Syria, stool samples from refugees arriving in Germany from Syria were tested for poliovirus, resulting in the identification of 12 individuals who excreted vaccine polioviruses [60].

In conclusion […….], (III) the flexibility to analyze additional samples in challenging situations, such as increased migration from countries struggling with poliovirus outbreaks or weakened health systems, is required."

Question 2): In results section: Why did the authors classify the patient in 4 age groups (>1 y, 1-4 y, 5-9 y and 10-14)?

Answer: Enteroviruses are predominantly detected in patients under 15 years of age. In order to make a more accurate evaluation and to point out different detection rates in certain age groups, we have subdivided this group of children and adolescents. Infants are particularly vulnerable. In recent years, the proportion of infants whose samples were sent in seemed to increase, while fewer samples were sent from older children. To observe these trends and possibly draw conclusions about the background (e.g. role of certain EV types in different age groups; data not shown) is one aim/one of the goals of EVSurv. The classification into infants, kindergarten age, school age, and middle school age seemed appropriate and has been used in other studies (DOI 10.1007/s00430-009-0133-6).

Sincerely,

Sabine Diedrich

Reviewer 3 Report

1) The definition of "Acute Flaccid Paralysis" first appears on line 34. The abbreviation "AFP" should be given here, not on line 53.

2) Line 53-54. Text: Rare complications of enterovirus infections are acute flaccid paralysis (AFP) or myelitis (AFM) ..." Since AFM is one of the diseases manifested by AFP, correction is recommended.

3) Line 108. The title "Performance/workflow of EVSurv". The abbreviation "EVSurv" should be decrypted.

4) Materials and Methods. The authors did not describe the statistical methods used to analyze the investigations. A section on statistical methods as well a section on ethical considerations should be included. 

5) Table 1. It is not clear why the first line (2006) is bold and underlined. The average number of samples enterovirus positive and typed  (columns 4 and 5 is better specified in the table. 

6) Figure 5. Not all of the legend is shown in the figure: what does red, light gray mean?

Author Response

Dear Reviewer,

Thank you for reading our manuscript carefully and for your positive and constructive assessment. We have addressed your very helpful comments as follows:

Point 1) The definition of "Acute Flaccid Paralysis" first appears on line 34. The abbreviation "AFP" should be given here, not on line 53.

Thank you. We introduced the abbreviation in Line 34 as you suggested.

Point 2) Line 53-54. Text: Rare complications of enterovirus infections are acute flaccid paralysis (AFP) or myelitis (AFM) ..." Since AFM is one of the diseases manifested by AFP, correction is recommended.

Answer: Thanks for the advice and excuse this inaccuracy. We have adjusted the wording as follows:  A rare complication of enterovirus infections is acute flaccid paralysis which may also result from acute flaccid myelitis (AFM) and poses a public health threat.

Point 3) Line 108. The title "Performance/workflow of EVSurv". The abbreviation "EVSurv" should be decrypted.

Answer: Thank you. We added the full term accordingly.

Point 4) Materials and Methods. The authors did not describe the statistical methods used to analyze the investigations. A section on statistical methods as well a section on ethical considerations should be included.

Answer: Thank you. You are right that this needs to be addressed. We added the following to the Material and Method Section (Line 100):

Prior to the launch of EVSurv it was proved by the Ethics Commission of the Medical Association of Lower Saxony that this is no medical research on humans with epidemiological and personal data. Therefore, an ethics vote was not required.

Data analysis was performed in Excel (percentages and means) and GraphPad PRISM (odds ratio).

Point 5) Table 1. It is not clear why the first line (2006) is bold and underlined. The average number of samples enterovirus positive and typed (columns 4 and 5 is better specified in the table.

Answer: Thank you very much. This seems to be a format problem we now corrected.

Point 6) Figure 5. Not all of the legend is shown in the figure: what does red, light gray mean?

Answer: Thank you very much. The legend is now completely displayed.

Sincerely,

Sabine Diedrich